

# Salinity tolerance and growth response of redroot pigweed (*Amaranthus retroflexus* L.): a comprehensive evaluation

Gohar Margaryan[1], Abhishek Singh[1], Vishnu D. Rajput[2], Mohamed Soliman Elshikh[3], Sapna Rawat[4] and Karen Ghazaryan[1]

[1] Faculty of Biology, Yerevan State University, Yerevan, Armenia
[2] Academy of Biology and Biotechnology, Southern Federal University Russia, Rostov-on-Don, Russia
[3] Department of Botany and Microbiology, College of Science, King Saud University, Riyadh, Saudi Arabia
[4] Department of Botany, University of Delhi, New Delhi, India

## ABSTRACT

Amaranths (*Amaranthus* L.) are multi-use crop species known for their nutritional quality and tolerance to salinity stress. As soil salinity has become an increasing issue globally, we conducted a study to test the salinity tolerance of one cultivar of *Amaranthus retroflexus* L. (Redroot pigweed). The plants were grown for 30 days in pot culture conditions at different salinity levels: non-saline, slightly saline, moderately saline, highly saline, and extremely saline (using two different rates: *extreme₁* and *extreme₂*. We assessed various growth parameters including plant height, stem diameter, root, stem and leaves fresh (FW) and dry weight (DW), water content (WC), photosynthesis rate ($P_n$), transpiration rate (E), water use efficiency (WUE), chlorophyll content index (CCI), nutrient content, phytodesalination potential, salt tolerance index (STI), and vegetation indices. Our findings indicated that at slight to moderate levels of salinity, growth parameters and other factors, such as STI and vegetation indices, were less affected compared to higher levels of salinity for *A. retroflexus* L.

## INTRODUCTION

Nowadays, salinity stress is one of the most significant abiotic stresses that affects crop production. of agricultural Worldwide, salt stress is projected to impact large area of soil lands (*Wander et al., 2019*). In the dry regions of Asia, Africa, South America, and Australia, where precipitations are insufficient to drain excessive salts, soil salinization—caused by the buildup of soluble salts *e.g.*, primarily NaCl and $Na_2SO_4$ in the soil's resulting upper horizon—is becoming an increasing constraint on agronomic practices for food production (*Butcher et al., 2016*). Sodic soils have an exchangeable sodium percentage (ESP) >6% and saline soils have an electrical conductivity >4 dS m$^{-1}$ (*Hassani, Azapagic & Shokri, 2020*). Natural salinization processes include erosion of salt rocks, bedrock, and air deposition; anthropogenic salinization processes are known as secondary salinization. When considering to the second scenario, saline or sodic soils might be caused by improper

Corresponding authors
Abhishek Singh, sinxabishik@ysu.am
Karen Ghazaryan, kghazaryan@ysu.am

irrigation methods (such as using brackish water), an increase in sea levels, or an over application or mismanagement of mineral fertilizers (*Eswar, Karuppusamy & Chellamuthu, 2021*). Topsoils contaminated with salt (>400 Mha, 0–30 cm) are predominantly saline (85%), with only a small percentage being sodic or saline-sodic (15%) (*Eswar, Karuppusamy & Chellamuthu, 2021*). Arid and semi-arid regions, as well as irrigated agricultural land, are experiencing a significant increase in salt-affected soils due to climate change (*Eswar, Karuppusamy & Chellamuthu, 2021*). The osmotic and ionic phases are the two main stages of salt stress in plants (*Singh et al., 2023*). The first cause of osmotic stress is reduced water intake caused by saline soils driven by a lower hydric potential. Osmotic stress directly affects plant growth and development (*Munns & Tester, 2008*). The ionic stress experienced by plants is the result of salt ions penetrating their tissues and competing with vital nutrients. Some species' tolerance methods include producing organic osmolytes to regulate osmotic pressure, whereas others depend on tissue partitioning and subcellular compartmentation to either exclude or accumulate salt (*Munns & Tester, 2008*; *Rajabi Dehnavi, Zahedi & Piernik, 2023*). The energy required by these tolerance mechanisms is a limiting factor in development and growth (*Munns et al., 2020*). Major agronomical crops are salt-sensitive which has encouraged the use of "indigenous" crops and wild crop relatives (WCR) for sustainable agriculture and food security (*Shan-e Ali Zaidi et al., 2019*; *Ye & Fan, 2021*; *Chourasia et al., 2021a*; *EL Sabagh et al., 2021*). The Amaranthus genus contains several *"indigenous"* crops and WCR, gaining attention for their salt tolerance (*Ye & Fan, 2021*).

The subcosmopolitan genus Amaranthus has 50–70 species of herbaceous plants, the majority of which are annuals belonging to the family *Amaranthaceae* (*Stetter & Schmid, 2017*). Plant species belonging to the *Amaranthaceae* family, specifically *Amaranthus* spp., are being studied as potential future crops because to their high mineral, protein, vitamin A, and vitamin C content, resilience to heat and drought, and accessibility to cultivation (*Rastogi & Shukla, 2013*). Beyond that, Amaranth seed oil contains natural bioactive compounds that could be used in pharmaceuticals (*Zhang, Kang & Che, 2019*). This is particularly accurate due to the high concentration of squalene in Amaranth, which is a key component in skin care products and penetration agents (*Hawrylak & Wolska-Mitaszko, 2007*). *Amaranthus* spp. are widely cultivated in temperate, subtropical, and tropical climate zones worldwide (*Ambika et al., 2002*). Amaranths are grown as a vegetable all over eastern Asia and the tropics (*Feine et al., 2019*). Various research on Amaranth show that these plants can tolerate severe drought and salt levels during their vegetative growth stage. It has been demonstrated that the stress response incorporates the reregulating ion of gas exchange, antioxidant defense, ion transporter modulation, and osmotic adjustment. The NAD malic-type $C_4$ photosynthetic pathway is expressed by Amaranths, giving them an advantage over $C_3$ plants in hot and dry climates due to their increased WUE (*Sage et al., 2007*).

Redroot pigweed, also known as *A. retroflexus*, is an annual weed plant species in the family *Amaranthaceae*. Among the most widespread dicotyledonous weeds in the world, *A. retroflexus* L. is found in many agricultural areas which causes a significant crop productionrelated problems (*Horak & Loughin, 2000*). Based on research by *Pearcy,*

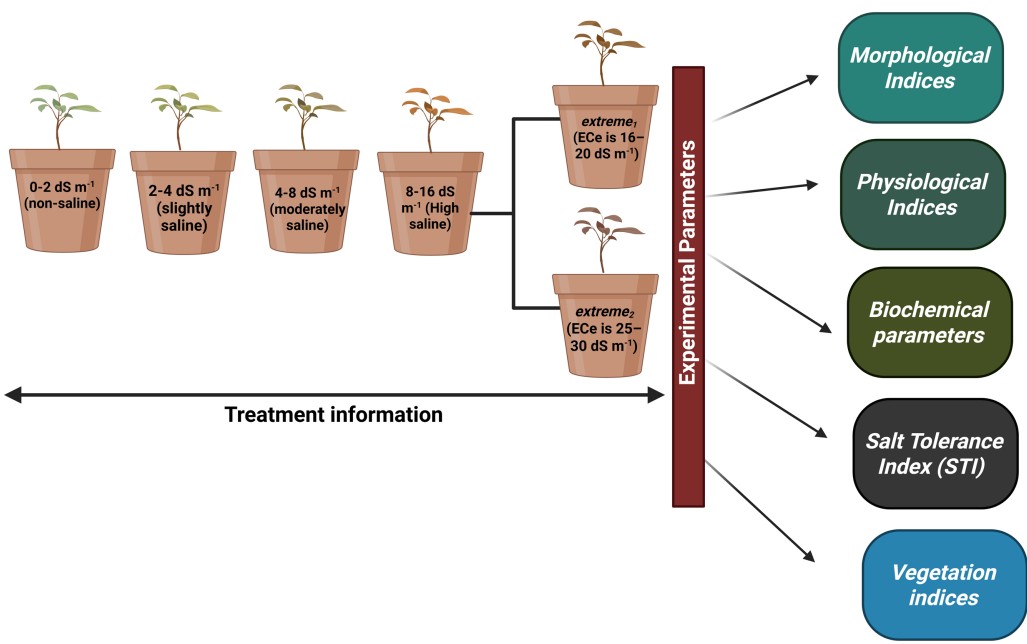

**Figure 1** **Full information about treatments plans and experimental parameters taken from plants.**

*Tumosa & Williams (1981)* suggested that *A. retroflexus* the is a $C_4$ plant that originally from North America but is currently it is found all over the globe (*Qaderi, Clements & Cavers, 2009*). One well-known survival mechanism that helps plants adapt to new environments is the capacity of *A. retroflexus* plants to change their morphological characteristics (*Gambino & Vilela, 2011*). Phenotypic plasticity allows *A. retroflexus* plants to undergo morphological changes under adverse conditions. The growth circumstances of an *A. retroflexus* plant are frequently associated with its morphological, physio-biochemical changes (*Mandák et al., 2011*). Redroot pigweeds are classified as weeds; however to date, little is known about the process by which *A. retroflexus* tolerates salt stress at the morphological and physio-biochemical levels. The aim of this study was to assess the resistance of *A. retroflexus* to salt stress and evaluate the extent to which these reactions are linked to the mechanisms involved in salt tolerance. For this purpose, we analyzed the effects of soil salinization ranging from non-saline to extremely saline on various parameters, including morphological, physiological, and biochemical. A screening of the salt tolerance of local *A. retroflexus* cultivars was carried out under pot culture conditions at non-saline, slightly saline, moderately saline, highly saline, and extremely saline using two different rates: *extreme*$_1$ and *extreme*$_2$ (Fig. 1). This allowed us to gain a deeper understanding of the salt response of *A. retroflexus* during the vegetative stage. We observed the growth and photosynthetic activity of the plants. $Na^+$ and $K^+$ concentrations were measured in the roots, stems, and leaves. As prolonged $Na^+$ accumulation affects mineral nutrition content, we measured all growth and physiological parameters related to salt stress.

## MATERIALS AND METHODS

### Plant materials and growth conditions

We used a homogeneous crop of young *A. retroflexus* growing on the Ararat Plain to collect seeds for our experiment. *A. retroflexus* seeds were surface-sterilized in 10% sodium hypochlorite for 15 min and then rinsed with distilled water many times. One kilogram of clay loam soil was used to fill pots. First, soil was deemed non-saline till soil electrical conductivity ($EC_e$) value was not reached <2.0 dS m$^{-1}$ (*Allison et al., 1954*). After 15 days, the seedlings were moved to plastic pots (20 L capacity with a height of 26 cm and a top diameter of 37 cm) with one plant per container. An experimental design, known as a complete random block (CRD), was employed, where each treatment represented five replicate pots. After the first leaf had fully matured, the plants were exposed to salt stress induced by various NaCl concentrations. The levels of salt stress were: non-saline, slightly saline, moderately saline, highly saline, and *extreme*$_1$ and *extreme*$_2$ correspondingly and full information of all treatments mentioned in Fig. 1. To prevent salt shock, NaCl was added slowly until the required amounts were reached. One seedling was placed in each of the five replicate containers that made up each treatment. Following a 25-day period of stable adaptation, measurements were taken every 10 days. For the duration of the experiment, non-perforated plastic containers were used to maintain a steady soil electrical conductivity and stop the leaching of salts.

### Morphological indices
#### Growth rate over time

Two important morphological characteristics, stem diameter measured with Vernier calipers and plant height, were used to assess the genotype of *A. retroflexus*. Plants from the salt and control groups were used in various combinations. To track the pace of growth over time, plant height was measured. Stem diameter was measured at the end of the pot experiment. Measurements were made of the plants grown in saline soils and compared to the plants cultivated in the control soil.

### Physiological indices
#### Biomass productivity

This study was the first to determine biomass based on the anatomical characteristics of *A. retroflexus* plants. Stems, leaves, and roots were pruned, and their fresh weight (FW) was measured. Then, the roots, leaves, and stems were cleaned, placed in paper bags, dried for 48 h at 70 °C in a lab oven, and weighed to calculate dry weight (DW) values. Six variables were measured: stem FW and DW, root FW and DW, after FW and DW of leaf.

#### Measurement of water content (WC) of roots, stems and leaves

The formula (*Repeta et al., 2016*) was utilized to determine WC for roots, stems and leaves:

$$WC = (FW - DW)/DW \tag{1}$$

where FW and DW are fresh and dry plant weight.

### Leaf succulence index (LS)

The LS was calculated as the total area and FW of 10 leaves of each plant according to *Agarie et al. (2007)* and *Jennings (1976)*:

$$LS = LFW/LA \tag{2}$$

where LFW is the leaf FW (g) and LA is the leaf area ($cm^2$).

### Chlorophyll content index (CCI)

It was determined that the CCl was a good measure of the stress condition of plants grown in soil that had been salinized. The CCI of the ten top leaves was measured using a CCM-200 plus chlorophyll content meter (*Margaryan et al., 2024*).

### Photosynthetic rate ($P_n$), transpiration rate (E) and water use efficiency (WUE)

Utilizing a portable photosynthesis system (CI-340), $P_n$ and E were measured under the following parameters: air temperature (25–30 °C), air pressure values (89.34–89.76 kPa), and air $CO_2$ concentration (430–475 $\mu$mol $mol^{-1}$). Following salt treatment for 10, 20, and 30 days, the measurements were completed. At nine to eleven in the morning, five young fully expanded leaves of each plant were used for all measurements. According to *Rabhi et al. (2012)*, the following formula was used to compute WUE, which is defined as net carbon uptake per amount of water lost from transpiring leaf area:

$$WUE = P_n/E. \tag{3}$$

## Biochemical parameters
### Uptake nutrient content, $K^+/Na^+$ ratio

In order to extract the ions, powdered dried roots, stems, and leaves were first digested in 0.5% $HNO_3$ solution (100 ml of solution per gram of sample) at 100 °C for 30 min. The filtrates were then quickly examined (*Al Hassan et al., 2016*). FP-I6431 flame photometer was used to measure the quantities of $Na^+$, $K^+$, and $Ca^{2+}$, whereas an I-160 M lab ionometer was used to measure the quantity of $Cl^-$.

### Relative electrolytic leakage (REL)

A stressed plant cell can be identified by the loss of electrolytes. This occurrence serves as "a parameter" for plant resistance to stress and is frequently used as a test to determine whether stress has damaged plant tissues (*Bajji, Kinet & Lutts, 2002*). 0.5 grams of the recently collected leaf tissue was combined with 50 mL of distilled water and allowed to ferment at room temperature in a test tube. Initial measurements of the electrical conductivity ($EC_1$) were obtained beforehand. After that, the samples were autoclaved for 15 min at 121 °C to eliminate any last ions from the tissue. The final electrical conductivity ($EC_2$) of the material was measured once it was cooled. The following is the calculation of the REL:

$$REL = (EC_1/EC_2) \times 100. \tag{4}$$

### Salt tolerance index (STI)

According to previous research the total plant DW (stem + root + leaf) in each pot grown with different salt concentrations was compared to the total plant DW produced under standard salt concentrations (*Tao et al., 2021*). STI was calculated based on this comparison. The STI results are shown as percentages. Salinity shoot toxicity (SST), shoot length stress (SLSI), and shoot weight stress (SWSI) indices were calculated using the following formula:

$$STI \text{ (\%)} = (\text{total DW at } S_x / \text{total DW at } S_1) \times 100, \tag{5}$$

where TDW is the total dry weight, $S_1$ is the control treatment, and $S_x$ is the x treatment.

$$SST \text{ (\%)} = (\text{shoot length of control} - \text{shoot length of treatment/shoot length of control}) \times 100 \tag{6}$$

$$SLSI \text{ (\%)} = (\text{shoot length of stressed plant/shoot length of control plants}) \times 100 \tag{7}$$

$$SWSI \text{ (\%)} = (\text{shoot weight of stressed plants/shoot weight of control plants}) \times 100 \tag{8}$$

## Vegetation indices

Vegetation indices are helpful in contemporary measurement approaches to assess plant health and are crucial for investigating (*Steddom et al., 2005*). In addition to identifying plant stress and diseases, vegetation indicators are useful for assessing the biophysical and biochemical characteristics of plants (*Rumpf et al., 2010*).

Spectrometers are one of the most often used tools for figuring out a plant's spectrum characteristics. These instruments are capable of measuring wavelengths between 300 and 1,300 nm in the field and lab (*Mac Arthur, MacLellan & Malthus, 2012*). According to Wasonga (*University of Helsinki, 2024*) one of the most widely used spectrometers is the *CI-710s SpectraVue Leaf Spectrometer*. It is intended to test a plant's ability to absorb, transmit, and reflect light across a broad wavelength range simultaneously. New functions added to the most recent SpectraVue Leaf Spectrometer include a touch screen, an integrated MS Windows operating system, a broad wavelength range (360–1,300 nm), and a leaf probe connection. As a result, data collection is streamlined, and the sensors are portable. The leaf spectrometer was equipped with two integrated wideband light sources (CID Bio-Science).

Meanwhile, the following spectral vegetation indices in *A. retroflexus* leaves were measured *in vivo* from 9 am to 12 am using a CI-710s Spectra Vue Leaf Spectrometer (CID Bio-Science) with compatible software: Anthocyanin Reflectance Index (ARI); Carotenoid Reflectance Index (CRI); Carter Index (Ctr); Structure Intensive Pigment Index (SIPI); Flavanol Reflectance Index (FRI); Triangular Vegetation Index (TVI); Greenness (G); Lichtenthaler Index (LIC); Gitelson and Merzlyak (GM1 and GM2); Zarco Tejada–Miller Index (ZMI); Normalized Difference Vegetation Index (NDVI); Normalized Phaeophytinization Index (NPQI); Normalized Pigment Chlorophyll Index (NPCI); Photochemical Reflectance Index (PRI); Simple Ratio Pigment Index (SRPI); Water Band Index (WBI); Plant Senescence Reflectance (PSRI); Transformed Chlorophyll

Absorption in Reflectance Index (TCARI); Vogelmann Red Edge Index (VREI); Red Edge Normalized Difference Vegetation Index (RENDVI); Chlorophyll Normalized Vegetation Index (CNDVI); Modified Red Edge Simple Ratio (MRESRI); Modified Chlorophyll Absorption Ratio Index (MCARI) and Modified Datt (MDATT) Index. The spectrometer on the leaf was calibrated before the measurements. The dried leaf was then placed into a leaf clip. Spectrometric measurements were performed on five leaves from each plant.

## Statistical analysis

The data were analyzed using SPSS-19 and Microsoft Excel 2021. Fisher's least significant difference (LSD) test was used to find statistically significant differences. Confidence intervals of 95% are shown as error bars in figures.

## RESULTS

### Morphological indices

Plant morphological traits, such as stem diameter and plant height, are important markers for determining the capacity of plants to tolerate salt stress. With increasing NaCl concentrations, plant height and stem diameter dramatically decreased. At extremely saline levels, a significant reduction in plant height was observed. In comparison to non-salinity levels, plant height was significantly decreased at $extreme_2$ salinity levels. More specifically, the maximum plant height was recorded at the non-salinity level, whereas a drop in plant height (by 54.8%) was noted at $extreme_2$ salinity levels. The trends for 10 daily measurements of plant height are presented in Fig. 2A. There was no discernible variation in average plant height up to moderate salinity levels. A stronger reduction was registered in plants under high to $extreme_2$ salinity levels, reaching a reduction of 1.4-fold for *A. retroflexus*. The reduced accessibility of nutrients and impaired cell development could be the cause of the reduction in plant length. Plants cultivated in salty soils first encounter osmotic stress, which restricts the absorption of water, and subsequently ion toxicity, which results in nutrient imbalance. The stem diameter of *A. retroflexus* was likewise affected by NaCl treatment. Over the course of the trial, the stem diameter data followed an identical trend to that of the plant height data. Up to $extreme_2$ salinity level, a declining trend in stem diameter was also noted (the difference being approximately 54.7% when compared to plants cultivated in non-saline soil) (Fig. 2B). Salinity stress generally decreased stem diameter and plant height; the extreme and non-salinity levels of the data were separated by a cluster of results. While the initial growth rates in the early weeks showed little difference, the differences increased throughout the remainder of the experiment.

### Physiological indices
#### Effect of salinity stress on A. retroflexus biomass productivity and water content

A considerable effect of different NaCl concentrations on plant biomass was observed. Increasing the salinity level from the first phase (non-salinity level) to the last phase ($extreme_2$ salinity level) reduced DW and FW gain in the roots, stems, and leaves of plants.

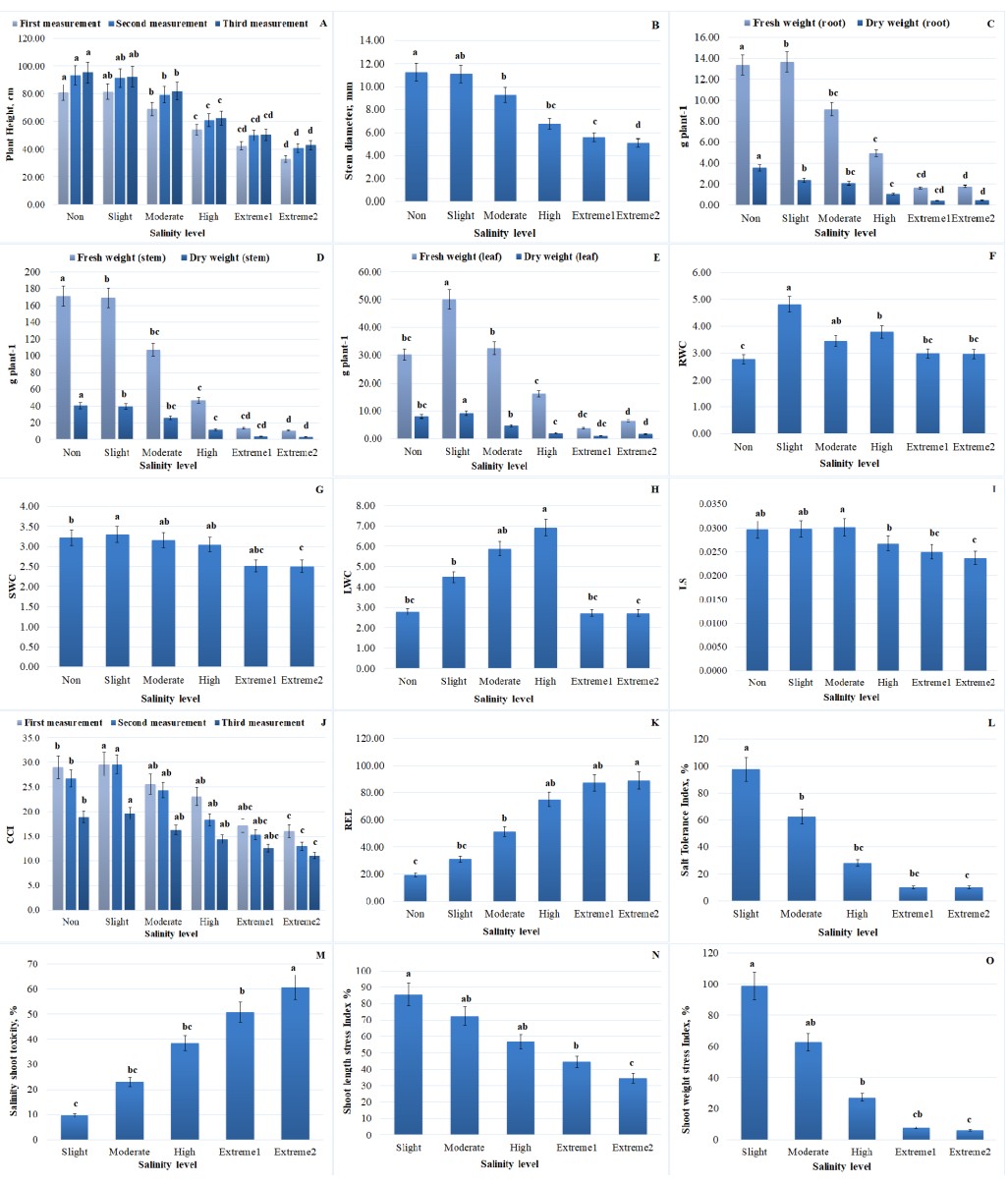

**Figure 2  Impact of salinity on several morphological and physiological traits of *A. retroflexus* during salt treatment.** (A) Plant height of plants, $n = 5$, $P < 0.05$, (B) stem diameter of plants, $n = 5$, $P < 0.05$, (C) FW and DW of root of plants, $n = 5$, $P < 0.05$, (D) FW and DW of stem of plants, $n = 5$, $P < 0.05$, (E) FW and DW of leaf of plants, $n = 5$, $P < 0.05$, (F) RWC of plants, $n = 5$, $P < 0.05$, (G) SWC of plants, $n = 5$, $P < 0.05$, (H) LWC of plant, $n = 5$, $P < 0.05$, (I) LSS of plants, $n = 10$, $P < 0.05$, (J) CCI of plants, $n = 50$, $P < 0.05$, (K) REL of plants, $n = 10$, $P < 0.05$, (L) STI of *A. retroflexus*, $n = 10$, $P < 0.05$, (M) SST of *A. retroflexus*, $n = 10$, $P < 0.05$, (N) SLSI of *A. retroflexus*, $n = 10$, $P < 0.05$, (O) SWSI of *A. retroflexus*, $n = 10$, $P < 0.05$.

The adverse concentration effects of salinity stress were clearly demonstrated by plants under high salinity levels. FW of roots, stems, and leaves of plants decreased by 86.6%, 93.7%, and 78.5%, respectively, and DW of roots, stems, and leaves decreased by 87.2%, 92.4%, and 78.2% under conditions of *extreme*₂ salinity level compared to plants grown in
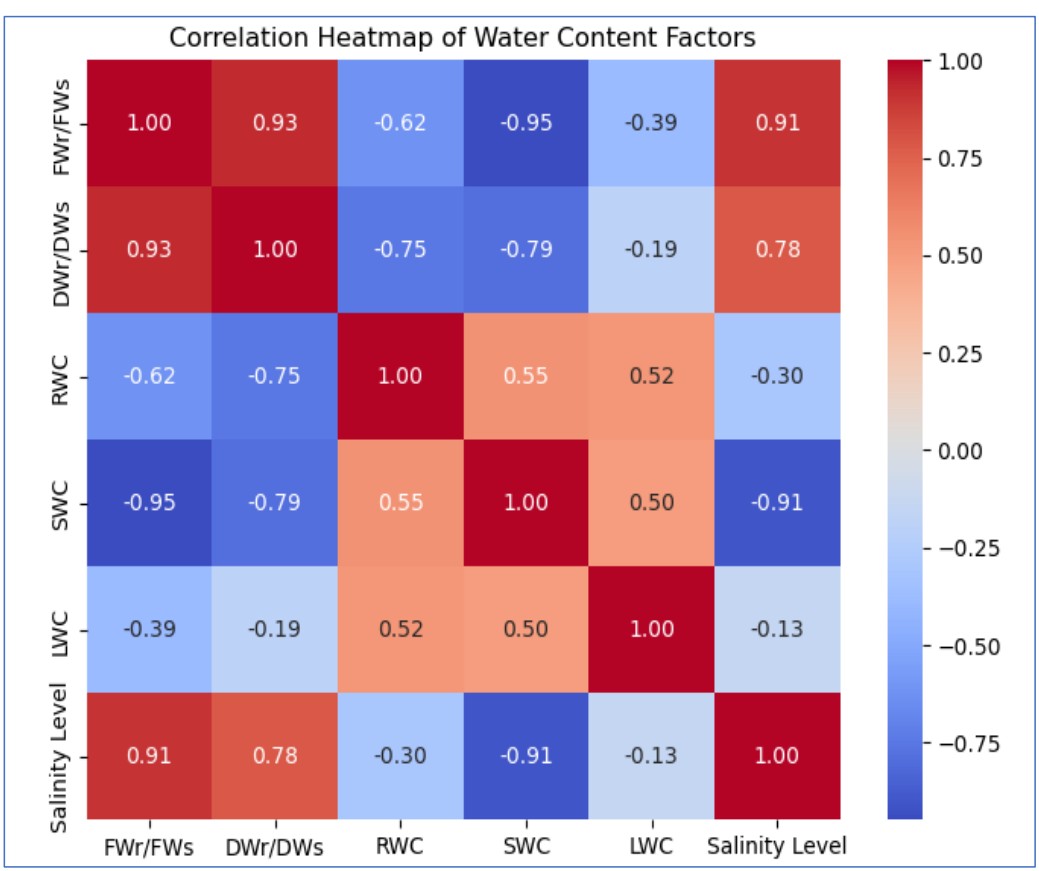

**Figure 3 The heatmap illustrates the correlation between salinity level and WC factors.** FWr/FWs and DWr/DWs show a strong positive correlation (0.93) with each other and increase with salinity (0.91, 0.78). SWC (−0.91) and RWC (−0.30) decrease with rising salinity, while LWC (−0.13) exhibits a weak negative correlation, suggesting minimal impact.

non-saline soil. Slightly salinized soil had the highest DW and FW values for *A. retroflexus* roots, stems, and leaves. Compared to the corresponding control, the statistical analysis showed a significant decrease in plant biomass. Comparative trends for roots (FW and DW), stems (FW and DW), and leaves (FW and DW) are shown in Figs. 2C–2E.

WC values for roots, stems, and leaves generally showed a tendency toward decline. When the value of WC dramatically dropped after the high salinity level, a tipping point was found. Following the high salinity level, the value of root, stem, and leaf WC greatly dropped. In comparison to plants growing in non-saline soil, stem water content (SWC) and leaf water content (LWC) of plants dropped by 22.1% and 2.1%, respectively, under high salinity conditions, while root water content (RWC) increased by 6.47%, respectively. Figures 3 and 4 illustrate the correlation between salinity level and WC factors.

$FW_r/FW_s$ and $DW_r/DW_s$ of *A. retroflexus* gradually increased with increasing soil salinity. The increase was steeper from moderate to *extreme$_2$* salinity levels (1.4-fold for FW and 1.2-fold for DW, respectively) compared to the control. $DW_r/DW_s$ and

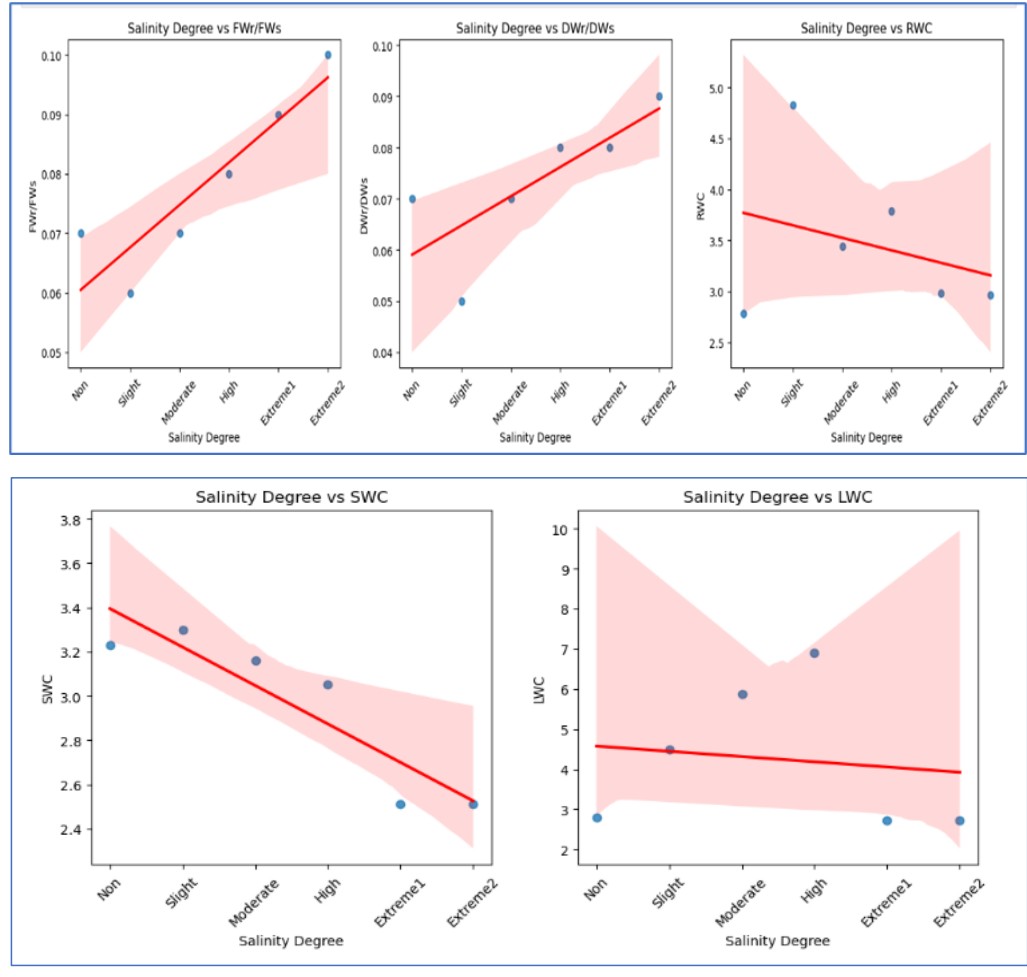

**Figure 4** **The scatter plots illustrate the correlation between salinity level and various WC factors.** FWr/FWs and DWr/DWs exhibit a negative correlation, decreasing with increasing salinity, indicating a decline in FW and DW ratios under salinity stress. In contrast, RWC and SWC show a positive correlation, suggesting that relative and soil WC increase with higher salinity levels. LWC has a weak positive correlation with salinity, indicating minimal influence on LWC.

$FW_r/FW_s$ decreased up to a moderate salinity level and then gradually increased. Figures 2F–2H and 3 depict the effect of salinity stress significantly affecting water distribution in plants. While root and shoot weight ratios increased, relative WC and declined, highlighting osmotic stress in response to salinity.

### Evaluation of water status of *A. retroflexus* under salinity stress

To assess *the water status of A. retroflexus*, the LS index was measured. At moderate salinity levels, the plants' LS index values peaked (Fig. 2I). The maximum variation was 20.2% when compared with plants cultivated in non-saline soils. As illustrated in Fig. 2I, the LS index declined proportionately as the salinity level increased. Nevertheless, a significant decline in succulence occurred once the high salinity level was reached.
### Evaluation of chlorophyll content

When compared to plants in the control group, CCI in plants damaged by salt decreased. As soil salinity increased, *A. retroflexus* exhibited a significant reduction in photosynthetic pigmentation (Fig. 2J). Three measurements were made during the vegetative period, and at each one, the value of CCI decreased. Furthermore, the decline became more pronounced as the salinity level increased. When the salinity level changed from high to $extreme_2$, there was a correspondingly considerable decline (the difference was 25.5%). In comparison to plants growing in non-saline soil, the value of CCI decreased by 41.5% under $extreme_2$ salinity level.

### $P_n$, E and WUE

These findings indicated that the $P_n$ value of *A. retroflexus* decreased (Fig. 5). This might be because throughout that time the plants developed a partial ability to adjust to the effects of the stress factor. An identical trend was also observed for the E values. In comparison to plants grown in non-saline soil, $P_n$ and E values in plants under $extreme_2$ salinity level generally dropped by 51.8% and 82.5%, respectively. Nevertheless, distinct WUE trends were noted throughout that time. More specifically, the plants started to use water more efficiently, and the WUE value increased by 2.7-fold as a result of improved tolerance to salinity stress and the development of more potent defense mechanisms against salt stress. Salinity stress significantly reduced $P_n$ and E across all measurements, indicating impaired gas exchange and reduced plant physiological performance (Fig. 5). While WUE showed an inconsistent trend, its increase in the third measurement suggests a possible compensatory mechanism under extreme salinity. Overall, higher salinity negatively affected plant gas exchange, likely leading to reduced growth and productivity.

## Biochemical parameters
### Uptake nutrient content, $K^+/Na^+$ ratio, REL

Increasing salinity stress caused distinct ion accumulation in the roots, stems, and leaves in diverse ways (Fig. 6). The findings indicate that, in comparison to control plants, a drop in the total $K^+$ concentration was seen in the roots and leaves of the plants ($K^+$ content decreased in roots and leaves 3.3 and 1.3 times, respectively). The opposite pattern was observed in the stems of plants; with increasing salinity level, there was an increase in total $K^+$ content compared to the control plants ($K^+$ content increased 1.0 times). Salinity enhanced $Na^+$ content in roots 6.6 times, stems 158.1 times and leaves 33.1 times, but decreased in the contents of $K^+$ and $K^+/Na^+$ ratio in roots (3.3 and 21.8 times, respectively), stems ($K^+/Na^+$ ratio 157.3 times) and leaves (1.3 and 45.0 times, respectively) in response to salinity. Although the increase in $Ca^{2+}$ content did not occur in such a dramatic manner (the content in roots and stems increased by 2.9 and 2.4 times, respectively, and decreased by 1.4 times in leaves), the content of $Cl^-$ in the roots, stems, and leaves was also significantly increased by 12.0-, 22.3 times and 24.6 times, respectively, along with the rise in salinity level. Salinity stress leads to a significant increase in $Na^+$ and $Cl^-$ accumulation, disrupting ionic homeostasis in plant tissues (Fig. 6). The sharp decline in $K^+$ and $K^+/Na^+$ ratio highlights severe potassium depletion, which may impair essential cellular functions. The mixed response of $Ca^{2+}$ suggests tissue-specific regulation, possibly playing a role in
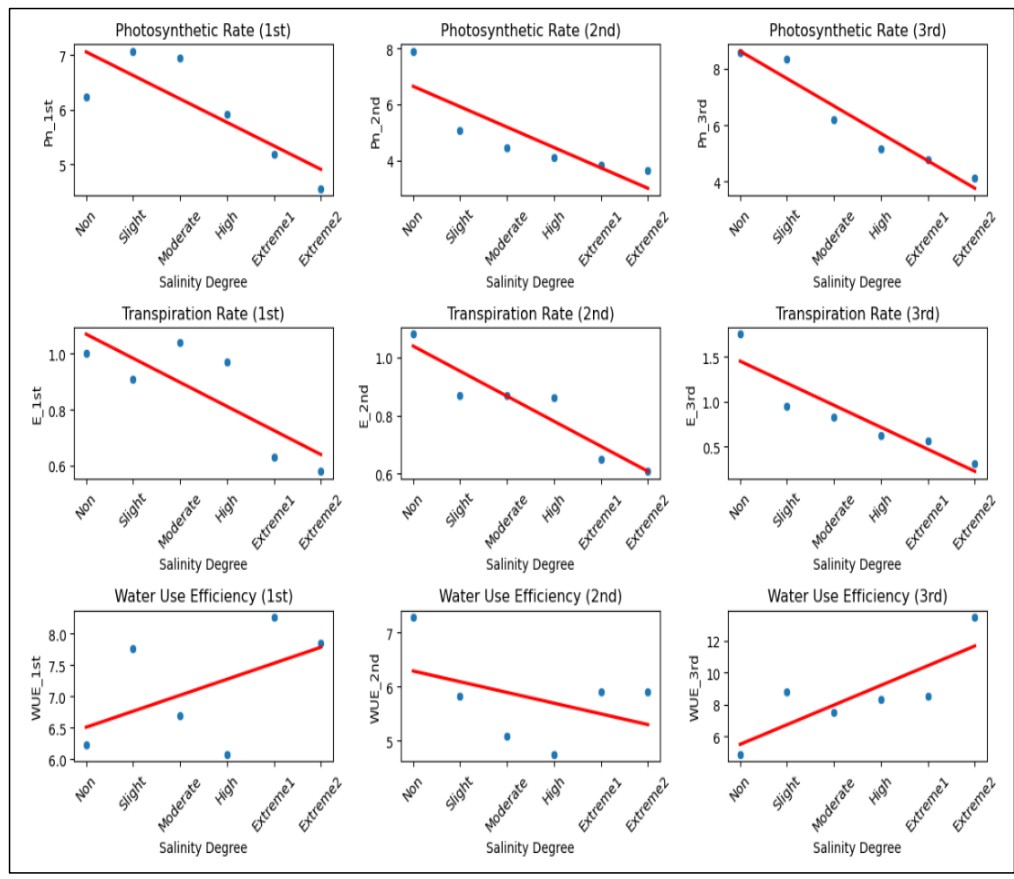

**Figure 5** The scatter plots illustrating the impact of salinity on $P_n$, E, and WUE across three measurements. $P_n$ decreases from 6.23 to 4.12, 7.90 to 3.63, and 8.57 to 4.12 $\mu$mol/m$^2$/s, while E declines from 1.00 to 0.58, 1.08 to 0.61, and 1.75 to 0.31 mmol/m$^2$/s as salinity increases. WUE fluctuates, decreasing from 6.23 to 7.84 (1st) and 7.28 to 5.91 (2nd), but increasing from 4.89 to 13.47 (3rd) under extreme salinity.

stress adaptation. Overall, high salinity induces ionic imbalance, potentially reducing plant growth and metabolic efficiency. Figure 6 depicting the correlation between ion content in plant tissues and salinity level.Salt stress also caused a considerable increase in REL value. While the rise was proportionate, there was a noticeable sharp shift in the REL value after the level of salinity reached a moderate one (Fig. 2K). When comparing non-saline soils to those with *extreme*$_2$ salinity, the value of REL rose by 4.5 times.

### Phytodesalination potential of A. retroflexus
The plants exhibited a considerable increase in Na$^+$ and Cl$^-$ accumulation in their roots, stems and leaves after 50 days of salt exposure, as compared to the control plants. It was found that by means of the stem of one plant 53.5 mg Na$^+$ and 54.1 mg Cl$^-$ can be removed from soils with moderate salinity level, 112.8 mg Na$^+$ and 113.0 mg Cl$^-$ under conditions of high salinity level, 72.5 mg Na$^+$ and 57.6 mg Cl$^-$ under conditions of

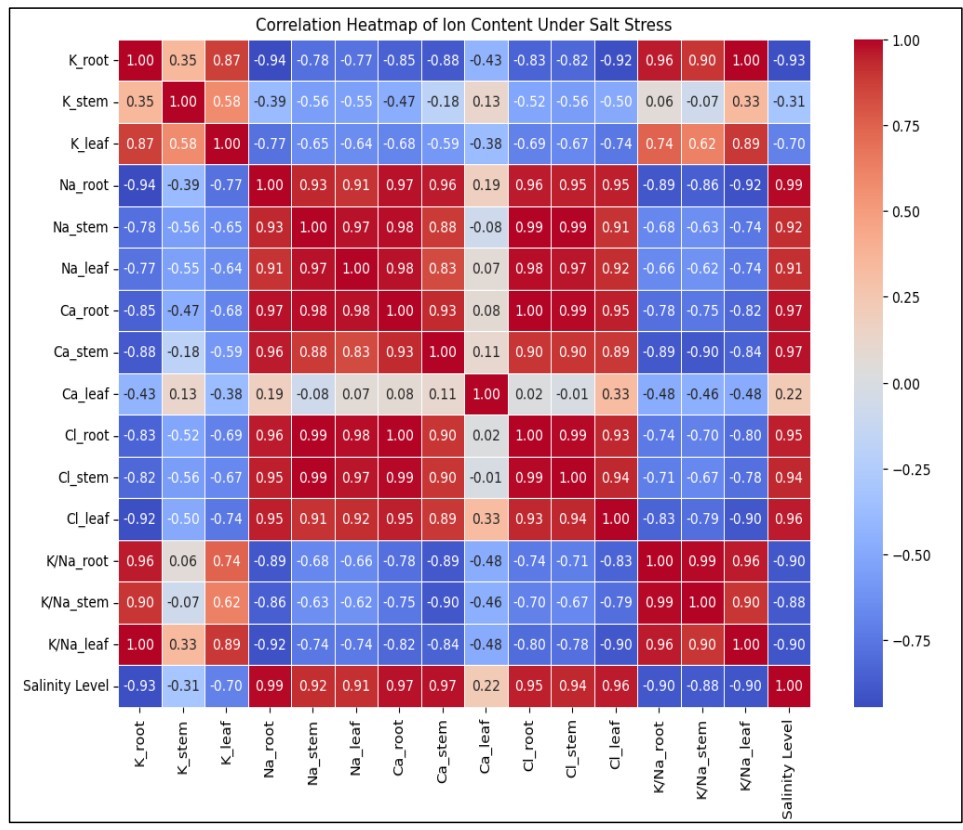

**Figure 6** **The heatmap depicting the correlation between ion content in plant tissues ($K^+$, $Na^+$, $Ca^{2+}$, $Cl^-$, $K^+/Na^+$ ratio) and salinity level.** $Na^+$ and $Cl^-$ levels strongly increase with salinity (0.97–0.99), while $K^+$ and $K^+/Na^+$ ratio show strong negative correlations (−0.93 to −1.00), indicating ionic imbalance under stress. $Ca^{2+}$ levels display mixed correlations depending on the tissue type, suggesting varied regulation in response to salinity.

*extreme*₁ salinity level, 105.7 mg $Na^+$ and 74.4 mg $Cl^-$ under conditions of *extreme*₂ level of salinity, respectively.

## Assessment of STI

The value of STI of plants showed the highest value at non-saline level. The variation was 89.7% when compared to plants cultivated in non-saline soil. As demonstrated in Fig. 2L, the STI value drastically dropped as salinity levels increased.

### SST, %

Regarding *A. retroflexus*'s SST, the opposite tendency was noted. With the increase in salinity levels the value of SST increased significantly (Fig. 2M). There was an 84.0% variation when compared to plants growing in non-saline soil.

### SLSI, %

The SLSI is a crucial metric for assessing *A. retroflexus* capacity for tolerance. As saline levels increased, the value of the SLSI dramatically dropped. The value of this index dramatically

decreased after it reached a moderate salinity level (Fig. 2N). In comparison, to plants cultivated in non-saline soil, there was a 59.5% variation.

### SWSI, %

Regarding the SWSI, a comparable pattern was noted (Fig. 2O). When salt levels rose, the SWSI value declined and the variation, when compared to plants cultivated in non-saline soil, was 93.6%.

## Evaluation of vegetation indices

Based on carotenoids and chlorophylls, the PRI and PSRI are commonly used to evaluate changes in the physiological status of plants. Five leaves from each plant were measured at different heights to calculate the PRI (Fig. 7L). The PRI normalizes the reflectance at 531 nm (R531) by combining it with a reference wavelength (R570) that is resistant to transient variations in the light energy conversion efficiency.

Throughout the vegetative period, the PSRI underwent substantial changes. Initially, the plants experienced less stress. It was found that when the PSRI falls below 0, it indicates the beginning of leaf senescence (Fig. 7K). Hence, the PSRI serves as a quantitative indicator of leaf senescence, while the photosynthetic efficiency is characterized by the photosynthesis rate (PRI). PSRI was found to be sensitive to carotenoids and the concentration ratio of chlorophyll.

Plants grown in saline and non-saline soils showed substantial differences in CRI2. Salinity stress can cause carotenoids to rise or fall according to the CRI2 data (Fig. 7AB). SIPI (Fig. 7I) of salt-affected plants (0.467) was significantly lower than that of the control plants (0.541) under NaCl treatment. Significant variations between salt-affected and control plants were identified under NaCl treatment, as demonstrated by the PRI (Fig. 7L), which followed the same trend as SIPI.

The NDVI is an index that is frequently and extensively used. NDVI is calculated as the ratio difference between reflectance in the red and near infrared bands. The NDVI (Fig. 7O) as greenness and disease indices of salt affected (0.456) experimental plants were significantly lower than the NDVI of non-affected (0.546) plants under NaCl treatment.

Significant differences in WBI (Fig. 7B) were exposed between control (1.034) and salt-affected (1.008) plants grown under NaCl treatment. The results of this investigation support the theory that salt stress causes WBI to decline. The WBI of salt-affected plants was significantly lower than that of non-affected plants because of reduced LWC. Pathogen inhibitors, such as flavanols and anthocyanins, were found in greater quantities shortly thereafter. The ARI1 and ARI2 of salt-affected plants maintained the same trend (Figs. 7AE and 7AF). Water scarcity is a result of decreased photosynthetic efficiency, as indicated by an increase in FRI (Fig. 7Y). Therefore, the FRI is inversely proportional to the WBI. Accordingly, it is reasonable to believe that the plants were under stress and forced to activate their defense mechanism, given that they lost a large amount of water and dramatically increased their flavanol content. It is well known that pigments such as xanthophyll, carotenoid, and chlorophyll are highly dependent on photosynthetic area and light. The consequence of this is a decrease in vegetation pigments. Therefore, the state of

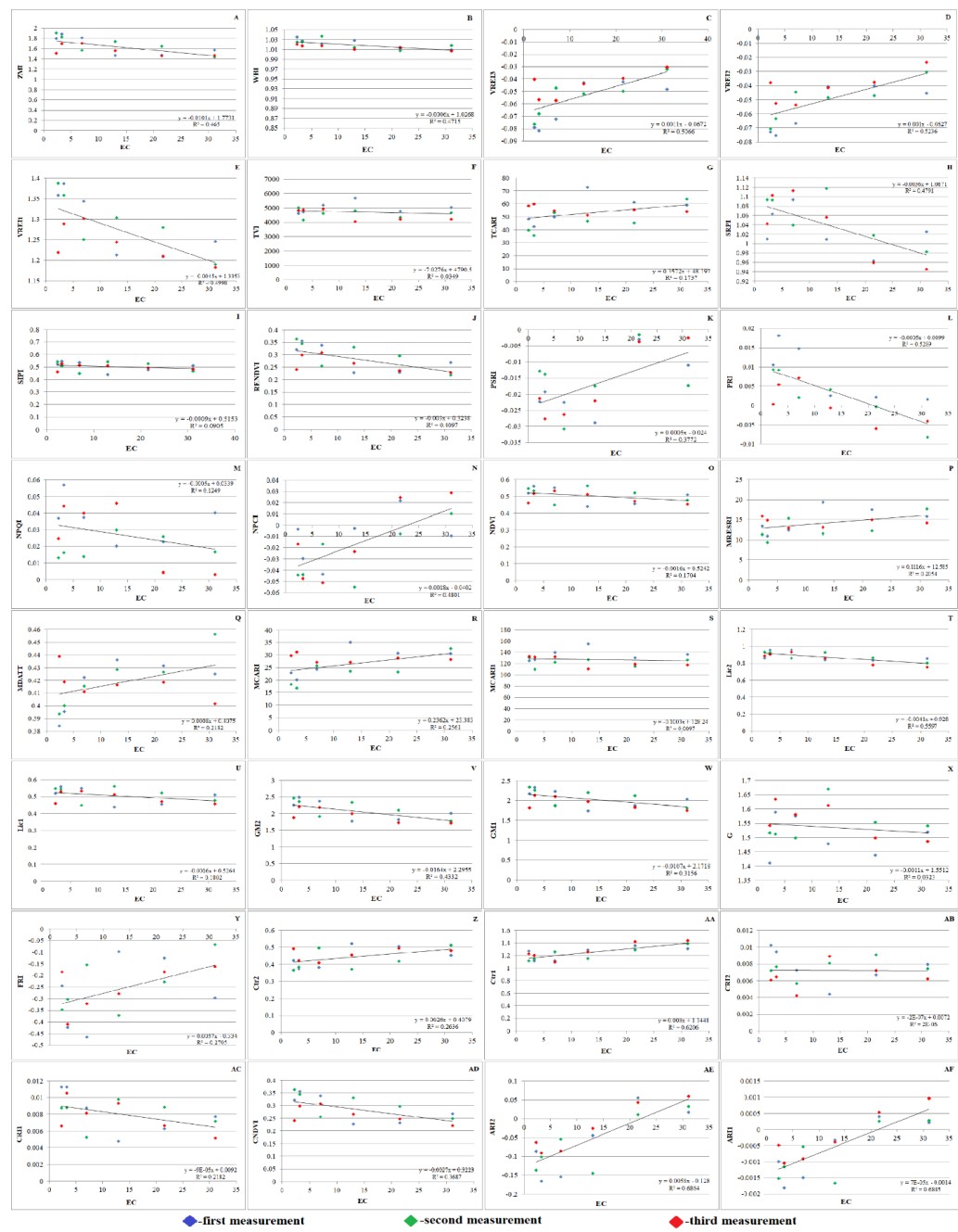

**Figure 7  Effect of salinity on some vegetation indices of *A. retroflexus* during salt treatment.** (A) Zarco Tejada–Miller Index (ZMI), $n = 5$, $P < 0.05$, (B) Water Band Index (WBI), $n = 5$, $P < 0.05$, (C) Vogelmann Red Edge Index3 (VREI3) $n = 5$, $P < 0.05$, (D) Vogelmann Red Edge Index2 (VREI2), $n = 5$, $P < 0.05$, (E) Vogelmann Red Edge Index 1 (VREI1), $n = 5$, $P < 0.05$, (F) Triangular Vegetation Index (TVI), $n = 5$, $P < 0.05$, (G) Transformed Chlorophyll Absorption in Reflectance Index (TCARI), $n = 5$, $P < 0.05$, (H) Simple Ratio Pigment Index (SRPI), $n = 5$, $P < 0.05$, (I) Structure Intensive Pigment Index (SIPI), $n = 5$, $P < 0.05$, (J) Red Edge Normalized Difference Vegetation Index (RENDVI), 

**Figure 7 (...continued)**
$n = 5$, $P < 0.05$, (K) Plant Senescence Reflectance Index (PSRI), $n = 5$, $P < 0.05$, (L) Photochemical Reflectance Index (PRI), $n = 5$, $P < 0.05$, (M) Normalized Phaeophytinization Index (NPQI), $n = 5$, $P < 0.05$, (N) Normalized Pigment Chlorophyll Index (NPCI), $n = 5$, $P < 0.05$, (O) Normalized Difference Vegetation Index (NDVI), $n = 5$, $P < 0.05$, (P) Modified Red Edge Simple Ratio (MRESRI), $n = 5$, $P < 0.05$, (Q) Modified Datt (MDATT) Index, $n = 5$, $P < 0.05$, (R) Modified Chlorophyll Absorption Ratio Index (MCARI), $n = 5$, $P < 0.05$, (S) Modified Chlorophyll Absorption Ratio Index1 (MCARI1), $n = 5$, $P < 0.05$, (T) Lichtenthaler Index 2 (LIC2), $n = 5$, $P < 0.05$, (U) Lichtenthaler Index 1 (LIC1), $n = 5$, $P < 0.05$, (V) Gitelson and Merzlyak Index 2 (GM2), $n = 5$, $P < 0.05$, (W) Gitelson and Merzlyak Index 1 (GM1), $n = 5$, $P < 0.05$, (X) Greenness (G), $n = 5$, $P < 0.05$, (Y) Flavanol Reflectance Index (FRI), $n = 5$, $P < 0.05$, (Z) Carter Index 2 (Ctr2), $n = 5$, $P < 0.05$, (AA) Carter Index 1 (Ctr1), $n = 5$, $P < 0.05$, (AB) Carotenoid Reflectance Index 2 (CRI2), $n = 5$, $P < 0.05$, (AC) Carotenoid Reflectance Index 1 (CRI1), $n = 5$, $P < 0.05$, (AD) Chlorophyll Normalized Vegetation Index (CNDVI), $n = 5$, $P < 0.05$, (AE) Anthocyanin Reflectance Index2 (ARI2), $n = 5$, $P < 0.05$, (AF) Anthocyanin Reflectance Index 1 (ARI1), $n = 5$, $P < 0.05$).

the host plant can be reliably determined using pigments and their decreasing or increasing indices.

## DISCUSSION

### Changes of plant and growth development characteristics of *A. retroflexus*

Our research findings suggested that *A. retroflexus* plants were more affected by higher salt treatments. This study examined the morphological, physiological, and biochemical changes in plant growth and development of the *A. retroflexus* plant genotype under different salt concentrations (non-saline, slightly saline, moderately saline, highly saline, and $extreme_1$ and $extreme_2$). As shown in Figs. 2A–2B, plant height and stem diameter were not significantly changed at slightly saline levels when compared to the control. However, these morphological traits decreased as salinity levels increased from moderately salinized to $extreme_2$ in *A. retroflexus* plants. Plant development and growth were affected by higher concentrations of NaCl, according to some preliminary research (*Grieve & Suarez, 1997*; *Wouyou et al., 2017*). Some reports also support our findings that higher NaCl concentrations decreased plant height and stem diameter in some plant species (*Zaman et al., 2020*; *Borsai et al., 2020*; *Pulvento, Houssemeddine Sellami & Lavini, 2022*). As an indicator of the plants' ability to survive under different abiotic stressors, plants'] FW and DW, biomass, and plant height indicate the plants' life-sustaining processes. The current study found that plants subjected to varying levels of salt stress (non-saline, slightly saline, moderately saline, highly saline, and $extreme_1$ and $extreme_2$) significantly reduced the biomass of roots, stems, and leaves (Figs. 2C–2E). Despite this, the biomass of *A. retroflexus* most drastically decreased at salinity levels ranging from high to $extreme_2$. According to the results, *A. retroflexus* can survive salinity levels of slightly salinized to moderately salinized without sacrificing their FW and DW of root, stem, and leaves, but these indicators were significantly stunted when exposed to salt levels of highly salinized to $extreme_2$, compared to the control condition. Various previous study also supports this finding which suggested that increasing the salinity stress decreased the plant biomass (*Butcher et al., 2016*; *Wouyou et al., 2017*; *Feine et al., 2019*; *Sarker & Oba, 2020*; *Singh et*
al., 2022b; *Singh et al., 2022a*; *Szymańska et al., 2022*; *Pulvento, Houssemeddine Sellami & Lavini, 2022*). Decline in plant biomass is also correlated with the water status in plants. A detrimental impact on plant-water interactions results from the buildup of ions in plant cells caused by fast ion absorption. Under conditions of salinity stress, plants experience a fall in turgor pressure and a decrease in WC within their cells because of an osmotic gradient caused by the soil's high salt concentration (*Ondrasek et al., 2022*). Our experiment finding suggested that Figs. 2F–2H, and 3 show the results of total water content in the roots, stems, and leaves of *A. retroflexus* plants affected at different levels of salinity stress. Salinity stress responsible for two types of stresses ionic and osmotic stress which reduced the photosynthesis rate resulting final carbon product of photosynthesis accumulation reduces and this process is responsible for buildup of biomass in plants (*Wouyou et al., 2017*; *Zeeshan et al., 2020*; *Tovar et al., 2020*; *Ondrasek et al., 2022*; *Liu et al., 2022a*; *Pulvento, Houssemeddine Sellami & Lavini, 2022*; *Balasubramaniam et al., 2023*; *Luyckx, Lutts & Quinet, 2023*; *Liu, An & Lai, 2024*). Thus, the FW and DW of *A. retroflexus* plants decreased. In response to severe salt stress, plant FW dropped dramatically, and this drop was expected as salinity increased (Figs. 2F–2H, 3–4). *Estrada et al. (2021)* discovered that the plant biomass matter content of *Amaranthus* was unaffected at 100 mM but higher salinity level decreased the total water content and biomass.

## Alteration in photosynthesis characteristics of *A. retroflexus* under salinity stress

Plants depend on photosynthesis, which produces both organic matter and energy that help in continuing plant growth and development. Inhibition of photosynthesis is the primary mechanism of salinity stress, which is responsible for the reduction in the formation of biomass in plants. One of the most essential photosynthetic pigments in plants is chlorophyll, which aids light absorption, transmission, and transformation. Tolerance levels in plants can be evaluated by analyzing their chlorophyll content (*Wang et al., 2022*). In this study, under salt stress, CCI decreased significantly more in the highly salinized to $extreme_2$ NaCl treatment compared to the slightly salinized to moderately salinized and control or non-saline treatments. When plants are exposed to salinity stress, their photosynthetic traits can serve as good indicators to analyses their salinity stress adoptability capacity (*An et al., 2021*). Plant gas exchange characteristics are known to be negatively affected by salinity stress which directly affected plant photosynthetic capacity (*Naz, Akram & Ashraf, 2016*; *Hnilickova et al., 2021*). This study found that *A. retroflexus* $P_n$, E, and WUE were reduced when exposed to salt stress. There was a greater reduction in $P_n$, E, and WUE under highly salinized to $extreme_2$ stress conditions compared to slightly to moderately salinized and non-saline levels. In the current study (Figs. 2J, 5) the levels of *A. retroflexus* $P_n$, E, and WUE were found to be significantly lower under situations of water deficiency or osmotic imbalance developed due to salinity stress (*Hawrylak & Wolska-Mitaszko, 2007*; *Wouyou et al., 2017*; *Feine et al., 2019*; *Zeeshan et al., 2020*; *Liu et al., 2022a*; *Pulvento, Houssemeddine Sellami & Lavini, 2022*; *Luyckx, Lutts & Quinet, 2023*; *Liu, An & Lai, 2024*). For a long time throughout the day, the stomata remain closed because of the water stress or osmotic stress caused by salinity stress. The results indicated that the inhibitory effect

on *A. retroflexus* photosynthesis was higher with highly salinized to *extreme*$_2$ salt stress compared to slightly to moderately salinized level. Due to stomatal closure under higher saline stress reduces $E$ value and intercellular carbon dioxide concentration due to this in our experiment the $P_n$ of *A. retroflexus* was too much lower under highly salinized to *extreme*$_2$ saline conditions compared slightly to moderate salinized level and control conditions (*Gandonou et al., 2018*; *Tarin et al., 2020*). Thus, stomatal factors mostly caused the reduction in photosynthesis of *A. retroflexus* under increased saline stress (highly salinized to *extreme*$_2$). Another key component of the restriction of photosynthesis caused by salt stress is the reduction in chlorophyll content (*Ondrasek et al., 2022*).

## Biochemical activity response of *A. retroflexus* against salt stress

During salinity stress biochemical changes occurs which induced osmotic and ionic stresses that cased ionic and osmotic imbalance into plants system but among these two ionic stresses is more lethal and altered the biological functions like membrane damage, inhibition of photosynthesis process, imbalance of ionic and osmotic conditions of plants (*Singh et al., 2022b*; *Singh et al., 2022a*; *Singh et al., 2023*; *Singh et al., 2024*). Plasma membranes are the primary ion-specific target of salt, which causes membrane damage and increases REL (Fig. 2K) (*Souri, Hatamian & Tesfamariam, 2019*). One of the most essential criteria for identifying salt-tolerant capacity of plants is to monitoring amount of electrolyte leakage from plasma membranes (*Ali & Ashraf, 2011*). In the current study, as the salt concentration increased, the REL value increased similarly across all treatments (Fig. 2K). Similar results were also observed in lettuce (*Lactuca sativa* L. cv. Orion), New Zealand spinach (*Tetragonia tetragonoides* (Pall) Kuntze) and common purslane (*Portulaca oleracea* L. cv. Green Purslane) (*Hniličková et al., 2019*). Ionic stress induced the nutrient-shortage or toxicity can hinder plant growth in deficient root zone nutrients. Low nutrient availability due to competition with main ions ($Na^+$ and $Cl^-$) makes mineral acquisition of nutrients challenging under salt-stress scenarios. This interaction mainly reduces $Ca^{2+}$ and $K^+$ like essential ions which participated in various biological function in plants (Fig. 6) (*Munns & Termaat, 1986*). The link between salt stress and nitrogen (N), phosphorus (P), and $K^+$ is complicated. N is a vital mineral in plant cells and increased $Cl^-$ absorption in saline environments can reduce shoot N uptake due to $Cl^-/NO_3^-$ antagonism (*Munns & Termaat, 1986*). Phosphorous is a key element which play important role in photosynthesis, transportation/storage, energy transfer in ETS system and less absorption of phosphorous is affected by salt stress. In soil with high levels of $Cl^-$ and $SO_4^{2-}$, phosphorous absorption may decrease due to high ionic strength and limited solubility of $Ca \pm P$ minerals. $Na^+$ and $K^+$ compete for root uptake sites resulting lowering $K^+$ and $Ca^{2+}$ (Fig. 6). $Na^+$-induced $K^+$ uptake reduction in plants is competitive, regardless of $Na^+$, $Cl^-$, or $SO_4^{2-}$ dominance (*Ali et al., 2013*). Salinity creates nutritional imbalances and lowers the yield of crops, according to extensive research on *Manihot esculenta* and *Zea mays* (*Ali & Ashraf, 2011*; *Ali et al., 2013*; *Shafiq et al., 2019*; *Razzaq et al., 2020*; *Ashraf et al., 2022*). Root, stem and leaves $Na^+$ and $K^+$ concentrations and $K^+/Na^+$ ratios play key markers that determine the salt tolerance capacity of plants. When compared to controls root, stem and leaves $Na^+$ and $K^+$ levels, as well as their $K^+/Na^+$ ratios, were significantly different under different levels

(non-saline, slightly saline, moderately saline, highly saline and *extreme*$_1$ and *extreme*$_2$) of salinity stress. A lower $K^+/Na^+$ ratio was observed for treatments with highly salinized to *extreme*$_2$ salt level, which explained the tolerance capacity of plants under these salinity levels. Previous many experiments also show that in different crop and *Amaranthus* plants including increasing the salinity levels decreasing the $K^+/Na^+$ ratio (*Zeeshan et al., 2020*; *Estrada et al., 2021*; *Chourasia et al., 2021b*; *Liu et al., 2022b*; *Luyckx, Lutts & Quinet, 2023*).

### Effect of salt stress on the tolerance activity and vegetation traits of *A. retroflexus*

*A. retroflexus* exhibits a wide range of morpho-physiological and biochemical reactions when exposed to salt and shows good tolerance at the initial stages of salinity. This study showed that all biochemical and morpho-physiological factors were linked to STI (Fig. 2L). As salinity stress increased, the STI value decreased (Fig. 2L). Certain STI parameters, such as SST (Fig. 2M) increased as salinity levels increased, while SLSI (Fig. 2N) and SWSI (Fig. 2O) decreased.

Plants absorb radiation slowly in the red band due to chlorophyll pigments, while responding quickly to near-infrared (NIR) wavelengths due to their strong radiation reflection. *Xue & Su (2017)* suggested that indices based on distinct spectral bands and the corresponding plant response might be used to assess the vegetation conditions. Various vegetation indices, such as the ARI2, CRI2, SIPI, FRI, NDVI, and G, are used to track plant growth and health (*Radočaj et al., 2023*). The SAVI characterizes green biomass (*Blackburn, 1998*). The PRI and PSRI are used to assess plant physiological status, with PRI normalizing reflectance at 531 nm and PSRI assessing light energy conversion efficiency. PSRI changes during vegetative periods, indicating leaf senescence. This quantitative indicator is sensitive to carotenoids and chlorophyll concentration ratio, and is a quantitative measure of photosynthetic efficiency, as characterized by the photosynthesis rate (PRI). Salinity stress affects plant growth, affecting carotenoids.

For leaf carotenoids, the SIPI is used. Salt-affected plants showed lower SIPI than control plants under the NaCl treatment. The NDVI, a vegetation index, is also used in studies on global environmental and climatic change (*Bhandari, Kumar & Singh, 2012*; *Gandhi et al., 2015*). The NDVI as greenness and disease indices of salt-affected plants were significantly lower than non-affected plants under NaCl treatment. These findings highlight the importance of understanding plant growth and adaptation to salinity stress. The current study found significant differences in WBI between control and salt-affected plants grown under NaCl treatment. This study supports the theory that salt stress causes WBI to decline, with salt-affected plants having lower WBI due to reduced LWC. Pathogen inhibitors, such as flavanols and anthocyanins, were found in greater quantities. Pigments such as xanthophyll, carotenoid, and chlorophyll are highly dependent on photosynthetic area and light, resulting in a decrease in vegetation pigments.

## CONCLUSION

Our experiment shows the adaptability and resilience of the *A. retroflexus* (redroot pigweed) species from our studies on how well it can tolerate different levels of salt. The results

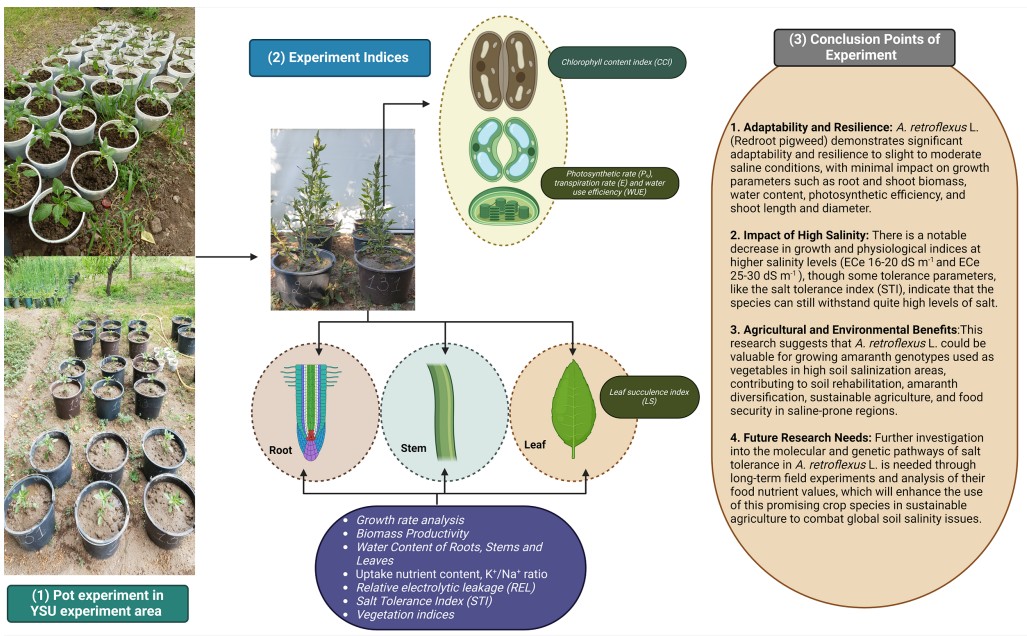

**Figure 8** Overall conclusion points for this research that show potential efficiency of *A. retroflexus* L. (Redroot pigweed) against salinity stress.

demonstrate that *A. retroflexus* may grow well in slight to moderate saline conditions with low impact on parameters including root and shoot biomass, WC, photosynthetic efficiency, plant height with stem diameter, and phytodesalination potential (Fig. 8). There were notable decreases in growth and physiological indices at higher to extreme salinity levels. Similarly, some tolerance parameters, such as STI, showed that it could withstand quite high levels of salt. This study showed that *A. retroflexus*. could provide valuable information for Amaranth genotypes that are used as vegetables for edible purposes and grow in higher soil salinization areas. The results also provide a broad range of information for soil rehabilitation and Amaranth diversification initiatives, which can be useful to make agriculture more sustainable and ensure food security in regions that are prone to higher salinity levels. Further investigation at the molecular and genetic pathways of salt tolerance in *A. retroflexus* L. is needed at the long-term field experiment level with analysis of their food nutrient values. To combat the growing problem of soil salinity on a global scale, these initiatives will increase the use of this promising crop species in sustainable agriculture.

## Funding

The Higher education and science committee MESCS Republic of Armenia provided the 23PostDoc-4D007 grant to support AS. The Higher education and science committee MESCS Republic of Armenia has provided grant numbers 21AG-4C075 to support KG. VDR is supported by the Strategic Academic Leadership Program of Southern Federal

University, known as "Priority 2030". The funders had no role in study design, data collection and analysis, decision to publish, or preparation of the manuscript.

### Grant Disclosures
The following grant information was disclosed by the authors:
Higher education and science committee MESCS Republic of Armenia: 23PostDoc-4D007, 21AG-4C075.
Strategic Academic Leadership Program of Southern Federal University (Priority 2030).

### Competing Interests
The authors declare there are no competing interests.

### Author Contributions
- Gohar Margaryan conceived and designed the experiments, performed the experiments, analyzed the data, prepared figures and/or tables, authored or reviewed drafts of the article, and approved the final draft.
- Abhishek Singh conceived and designed the experiments, performed the experiments, analyzed the data, prepared figures and/or tables, authored or reviewed drafts of the article, and approved the final draft.
- Vishnu D. Rajput conceived and designed the experiments, performed the experiments, analyzed the data, prepared figures and/or tables, authored or reviewed drafts of the article, and approved the final draft.
- Mohamed Soliman Elshikh conceived and designed the experiments, analyzed the data, prepared figures and/or tables, authored or reviewed drafts of the article, and approved the final draft.
- Sapna Rawat conceived and designed the experiments, analyzed the data, prepared figures and/or tables, authored or reviewed drafts of the article, and approved the final draft.
- Karen Ghazaryan conceived and designed the experiments, performed the experiments, analyzed the data, prepared figures and/or tables, authored or reviewed drafts of the article, and approved the final draft.

### Data Availability
Data are available in the Supplemental files.

### Supplemental Information
Supplemental information for this article can be found online at http://dx.doi.org/10.7717/peerj.19717#supplemental-information.

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
