# Peer review of "Salinity tolerance and growth response of redroot pigweed (Amaranthus retroflexus L.): a comprehensive evaluation"

_PeerJ, doi:10.7717/peerj.19717_

## Round 0.1 · original submission · Major Revisions

Abiotic stress caused by salinity is an important phenomenon we should find how crops like Amaranthus retroflexus respond to this adverse environmental condition. Your research involves valuable insights to understand it. Nevertheless, it is essential to address certain technical details to enhance the article further. I encourage you to carefully review the reviewers' suggestions and thoughtfully consider each recommendation. If you find yourself in disagreement with any specific suggestions, providing a clear and well-supported rationale for your viewpoint would be highly beneficial.

·

Basic reporting

-

Experimental design

-

Validity of the findings

-

Additional comments

Dear author, I reviewed your current study - quite interesting and within the scope of the journal needed, but I recommend a major revision for your paper:

In introduction section was written well, but in last needed to add a clear objective of the research.

For the material method section:
Was CRD in that this was repeated across multiple blocks, or was it simply six replicates per treatment of a single block?
Although it is noted that salt levels varied from non-saline to extreme,2 the precise NaCl concentrations for each treatment are not indicated in the main text. Can the authors please provide the NaCl concentrations (in mM or dS/m) at each salinity level?
How was soil electrical conductivity (ECe) maintained uniformly throughout the experiment, particularly in non-perforated pots? Are you periodically measuring EC, which confirms stable salinity levels maintained throughout, at least during the 25-day adaptation, and later?
Spectrometric measurements were made after calibration on dried leaves. In vivo measurements with CI-710s are usually performed on fresh leaves to represent real-time physiological status. Could the authors explain or justify this approach?
It looks like there are six biological replicates for each treatment. Could the authors confirm whether these were used for all physiological, biochemical, and spectrometric measurements? Were technical replicates done as well, and what statistical tests were used for analysis?
Were the plants grown in a controlled environment (e.g., greenhouse, open field)? Using these data (temperature, humidity, light intensity, and irrigation frequency) is important for the reproducibility and interpretation of the effects of salinity stress.

For the result section:
Please give more details on how salinity was established and maintained during the experiment, and what criteria were used to select the salinity gradient (non-saline to extreme 2). Was soil EC (electrical conductivity) measured routinely to ensure salinity was consistent and stable?

The study does report percent changes and fold differences at the level of salinity, but whether all of them are statistically significant? Can the authors please elaborate on any statistical tests used (e.g., ANOVA, post-hoc comparisons) and whether interactions between factors (e.g., organ type × salinity level) were evaluated?

The higher WUE under extreme salinity indicates compensation. Do the authors mean that this increase in WUE represents an adaptive advantage or potentially just less transpiration during a period of stress? What role might stomatal behaviour or internal CO₂ concentration play in the production of this trend?

These results indicate the disparity in ion accumulation (Na⁺, K⁺, Ca²⁺, Cl⁻) of plant tissue. Could the authors elaborate on possible tissue-specific sequestration (i.e., vacuolar compartmentalization, selective transporters) that might be involved in A. retroflexus salt tolerance or exclusion strategies?
The authors examined differences in spectral indices (PRI, PSRI, NDVI, WBI). How do you foresee applying these indices in field-based versus remote sensing platforms to monitor A. retroflexus performance under natural salinity stress, and/or how could your approach contribute to species distribution modeling? Would these indices be able to discriminate between salt stress and other environmental aspects (drought, etc.)?

The discussion section needed to include a few more information regarding 2020-2025 that directly relate to the current study, and the conclusion section needed to be rewritten with the importance of current findings with the future perspective of the current study.

Reviewer 2 ·

Basic reporting

I reviewed this paper, which falls within the scope of the journal with recent analysis and scientific trends, but I have a few major suggestions to revise the paper, which would enhance the overall quality of the manuscript:

Experimental design

Authors should provide clearer reasons for why NaCl concentrations were used, particularly for the two extreme levels of salt added (extreme1 and extreme2).

Statistical Analysis: The Authors stated that SPSS and LSD tests were used, however, it is recommended to mention p-values, effect sizes, and mean separations to justify the statistical interpretation.

Nutrient Analyses: Flame photometry for Na+ and K+ is fine, but details of a calibration curve, standards, and accuracy of measurements should be provided.
Vegetation Indices: Description of why some indices (PRI, PSRI, etc.) are especially relevant to the salinity tolerance of Aretio is needed. These indices were chosen among the many that are available for analysis.

Chlorophyll: There is no reference to validation of the values measured by the CCM-200+ device with a laboratory method of chlorophyll extraction. Please discuss.
7.2. Biochemical Parameters Relative Electrolytic Leakage (REL) was observed, but the estimation of lipid peroxidation (for example, MDA content) could increase the understanding of the physiological damage.

Photosynthesis Measurement: Can you tell whether the light intensity (PPFD) was uniform when photosynthesis (Pn) was measured, and whether comparability existed?

Water Use Efficiency (WUE): The WUE patterns at high salt levels are unusual. Please elaborate on whether this may relate to changes in carbon allocation or stress physiology.

Interpretation of K⁺/Na⁺ Ratios: The section on ionic balance might benefit from the inclusion of more recent reviews on plant ion homeostasis under salinity.

Importance of Phytodesalination: The concept of ‘phytodesalination potential’ is new in the present study. Nevertheless, comparison of A. retroflexus with known halophytes would be very relevant.

Validity of the findings

Figures and Tables: Some figures are cited (e.g., Fig. 2 A-B) but not included. Continue embedding and numbering figures and figure legends accordingly.

Consistency of Units: The use of “mg” or “g” as units should be applied uniformly according to the SI requirements, and the unit should be in full or in the abbreviation.

Title Specificity: When the authors refer to a salinity, perhaps an emphasis on the type of salinity tested (e.g., "NaCl induced Salinity Stress") could be included in the title, as the work done seems directed at breaching its use in greenhouses based on cost.

Abstract: It would be preferable to include a minimal description of these key results with numbers (i.e., % decrease in biomass at extreme salinity) in the abstract.

Introduction Flow: The introduction is a bit bogged down with 'salty' general information. Turn your attention faster to Amaranthus retroflexus and why it was chosen.

Materials and methods. The Materials and Methods section. Some of the subchapters there (e.g., "Leaf Succulence Index") may not be visible directly due to formatting. The same style of heading should be used for all levels in the Materials and Methods section.

Structure of the present Discussion: The Current Discussion rehashes certain Results far too much. Try to concentrate by comparing your results to prior studies.

Conclusion: The Conclusion is OK but could be better closed with 1–2 sentences stressing the possible application of A. retroflexus as a crop in salt agriculture.

References: Some references contain a formatting problem (e.g., “CSL STYLE ERROR: reference with no printed form.”). Kindly verify/rephrase them following the journal instructions.

Additional comments

Language polishing: The manuscript requires a suitable English language editing for its grammar, word choice, and sentence structure, to enhance readability (e.g., “adoption process” instead of “adaptation process”).

Reviewer 3 ·

Basic reporting

see PDF file

Experimental design

see PDF file

Validity of the findings

see PDF file

Additional comments

see PDF file

Annotated reviews are not available for download in order to protect the identity of reviewers who chose to remain anonymous.

---

## Round 0.2 · accepted · Accept

I would like to thank you for accepting the referees' suggestions and improving your article based on their suggestions. Your article is ready to publish. We look forward to your next article.

E.g. remove "comprehensive" from the title.

·

Basic reporting

This article can be accepted aa it resolved all my concerns.

Experimental design

Correct

Validity of the findings

Correct

Additional comments

No additional comments.